# High-Throughput/High Content Imaging Screen Identifies Novel Small Molecule Inhibitors and Immunoproteasomes as Therapeutic Targets for Chordoma

**DOI:** 10.3390/pharmaceutics15041274

**Published:** 2023-04-18

**Authors:** Amrendra K. Ajay, Philip Chu, Poojan Patel, Christa Deban, Chitran Roychowdhury, Radhika Heda, Ahmad Halawi, Anis Saad, Nour Younis, Hao Zhang, Xiuju Jiang, Mahmoud Nasr, Li-Li Hsiao, Gang Lin, Jamil R. Azzi

**Affiliations:** 1Transplant Research Centre, Division of Renal Medicine, Department of Medicine, Brigham and Women’s Hospital, Boston, MA 02115, USArheda@bwh.harvard.edu (R.H.);; 2Department of Medicine, Harvard Medical School, Boston, MA 02115, USA; 3Department of Microbiology and Immunology, Weill Cornell Medicine, New York, NY 10065, USA

**Keywords:** chordoma, proteasome, high-throughput screening, rare disease, cell proliferation

## Abstract

Chordomas account for approximately 1–4% of all malignant bone tumors and 20% of primary tumors of the spinal column. It is a rare disease, with an incidence estimated to be approximately 1 per 1,000,000 people. The underlying causative mechanism of chordoma is unknown, which makes it challenging to treat. Chordomas have been linked to the T-box transcription factor T (TBXT) gene located on chromosome 6. The TBXT gene encodes a protein transcription factor TBXT, or brachyury homolog. Currently, there is no approved targeted therapy for chordoma. Here, we performed a small molecule screening to identify small chemical molecules and therapeutic targets for treating chordoma. We screened 3730 unique compounds and selected 50 potential hits. The top three hits were Ribociclib, Ingenol-3-angelate, and Duvelisib. Among the top 10 hits, we found a novel class of small molecules, including proteasomal inhibitors, as promising molecules that reduce the proliferation of human chordoma cells. Furthermore, we discovered that proteasomal subunits PSMB5 and PSMB8 are increased in human chordoma cell lines U-CH1 and U-CH2, confirming that the proteasome may serve as a molecular target whose specific inhibition may lead to better therapeutic strategies for chordoma.

## 1. Introduction

Chordoma is a rare primary bone cancer that affects mainly the skull base, mobile spine, and sacrum [1,2,3]. In the United States, about 300 new chordoma cases are diagnosed yearly. Under normal circumstances, embryonic notochord cells express a transcription factor called brachyury, which is duplicated in patients with chordoma, suggesting a possible role in the pathogenesis [4,5]. The diagnosis of chordoma is challenged by chondrosarcoma, another mesenchymal tumor, due to their similar characteristics and cytokeratin staining and S100 protein. Nevertheless, recent data suggest brachyury as an essential biomarker for chordoma that can assist in diagnosis when used with cytokeratin staining [6].

In most cases, surgical removal followed by radiation therapy is the best long-term option [7], but due to bone invasion, complete tumor removal is impossible. Additionally, limited radiation doses can be applied due to its adjacent location to delicate brain structures, such as cranial nerves and the brain stem. Even after surgery and/or radiation, chordomas can relapse, leading to multiple surgeries over several years. Tumor metastasis to the lungs, liver, bones, and skin occurs in 20–40% of patients with chordomas of the spine [3,8]. Chordomas are malignant and potentially life-threatening tumors with a median survival of 7 years.

Conventional chemotherapies failed to show significant benefits to patients with chordoma. Other target chemotherapy agents, such as inhibitors of the signal transducer and activator of transcription 3 (STAT3), epidermal growth factor receptor (EGFR), platelet-derived growth factor receptor (PDGFR), and PI3Kinase alone or in combination, are still under investigation [9,10,11].

The first preclinical studies on chordoma drug screening were limited to two validated human chordoma cell lines, U-CH1 and U-CH2, which show brachyury expression, a characteristic marker for chordoma. One study showed that high-throughput screening (qHTS) identified an essential inhibitory role of Bortezomib against U-CH1 and U-CH2 cells [12]. Recently, another small molecule, THZ1, was also identified, targeting the chordoma brachyury transcription factor [13]. Erlotinib reduces the growth of patient-derived chordoma xenografts [14]. The phase I clinical trial of the linsitinib/erlotinib combination was tolerable with preliminary evidence of activity, including durable responses in cases unlikely to respond to the erlotinib monotherapy [15].

Immunostaining for PDGFRα, EGFR, and c-MET in the chordoma patients’ samples showed an increase in these three proteins [16] and prompted multiple investigations to target EGFR. EGFR inhibitors, identified as a potential therapeutic target for chordoma by screening 1097 focused libraries of compounds [17], and specific inhibitors that target EGFR and cyclin G-associated kinase (GAK) showed efficacy in chordoma cells [18]. Another EGFR inhibitor, Afatinib, showed promising results against chordoma cells [19]. Additionally, 4-anilinoquinolines and 4-anilinoquinazolines were developed for targeting the water network of EFGR, and showed efficacious results in the chordoma [20].

Given the failure of conventional therapy in improving chordoma’s prognosis, we conducted a cell-based high-throughput/high content imaging screen on human chordoma cell line U-CH1. The aim of this study is to find a small molecule inhibitor for the growth inhibition of chordoma cells. Finding the small molecule inhibitors may be tested for clinical use on chordoma patients following its confirmation in mouse models.

## 2. Materials and Methods

### 2.1. Chemicals and Cells

Hoechst 33342 stains were purchased from Life Technologies (Cat no. H3570). Human Chordoma cell lines U-CH1 (Cat. no. CRL-3217) and U-CH2 (Cat. no. CRL-3218) were purchased from ATCC (Manassas, VA, USA). Cells were grown in Iscove’s Modified Dulbecco’s Medium (Cat. no. 30-2005, ATCC): RPMI-1640 Medium (Cat. no. 30-2001, ATCC) (4:1), 10% FBS (Cat. no. 30-2020, ATCC) supplemented with 2 mM L-glutamine (Cat. no. 30-2214, ATCC). HEK-293 (Cat. no. CRL-1573) and HUVEC (Cat. no. PCS-100-013) cells were purchased from ATCC and were grown in Eagle’s Minimum Essential Medium (Cat. no. 30-2003, ATCC) supplemented with 10% fetal bovine serum, and Vascular Cell Basal Medium (Cat. no. PCS-100-030, ATCC) supplemented with Endothelial Cell Growth Kit-BBE (Cat. no. PCS-100-040, ATCC), respectively. PSMB5 (Cat. no. 11903), PSMB8 (Cat. no. 13635), and β-Actin (Cat. no. 3700) antibodies were purchased from Cell Signaling Technology (Danvers, MA, USA). Doxorubicin was purchased from Sigma (Cat. no. D1515). Compound libraries were provided by the Institute of Chemistry and Cell Biology (ICCB) Longwood, Harvard Medical School, Boston, MA, USA.

### 2.2. Primary Screening

We developed a cell-based counting method using Hoechst 33342, a nuclear stain dye. One thousand cells in 30 µL medium were plated using a Multidrop Combi Reagent Dispenser (Thermo Scientific, Waltham, MA, USA) in a 384-well plate and were incubated for 24 h at 37 °C. The first column of wells was left without the addition of any compound. One hundred nanoliters of DMSO, as vehicle control, was added to the 24th column, and doxorubicin positive control (5 µM) was added to the 2nd column. One hundred nanoliter test compounds were added to columns 3–22 from the library plate using the Epson E2C2515-UL Scara robot (Epson, Suwa, Tokyo). The compound concentrations were 3.3 µM for most plates, with some exceptions where compounds were used at 0.33 µM. The primary assay was performed in duplicate on two separate plates. Following 72h incubation at 37 °C in a humidified CO_2_ incubator, cells were fixed by adding 4% paraformaldehyde for 30 min and were then washed thrice with PBST (0.1% Tween 20). After washing, 0.1 µg/mL of Hoechst 33342 was added and incubated for 30 min. Cells were washed twice with PBST and once with PBS. Thirty µL of PBS containing 0.01% Sodium azide was added, and the plate was imaged using IXM-C version 6.7.2.290 with 4X objective. Images were analyzed using IXM-C software, and nuclear (number of Hoechst-positive nuclei) were counted.

### 2.3. Z′-Factor and Z-Score

The higher the Z-prime factor, the higher the reproducibility and robustness of the assay. We optimized our assay to obtain a significantly high and robust Z-prime factor, calculated by inhibiting cell growth by counting Hoechst-positive nuclei using Doxorubicin (positive control) compared to DMSO alone (negative control) treated cells. We utilized the formula Z′ = 1 − (3σ_p_ + 3σ_n_)/(μ_p_ − μ_n_), where σ is the standard deviation, μ is the mean, p is a positive control, and n is the negative control [21]. The Z-score for each compound was calculated by its potential to reduce cell proliferation and lower standard deviation between the replicate plates.

### 2.4. Re Confirmation and Dose Dependency Studies

Small volumes (1.5 µL) of potential hits from the primary screen were cherry-picked using a 384-well plate with a robotic dispenser for dose dependency studies. The dose dependency studies were performed in the U-CH1 cell line: One thousand cells were plated in a 384-well plate using a Multidrop Combi Reagent Dispenser and incubated for 24 h. The highest dose used was the same at which the primary screening was performed in duplicate wells. Eight doses from 3.3 μM or 0.33 μM in duplicate of 1:3 dilutions were added by an HPD300 (Hewlett Packard, Palo Alto, CA, USA), and DMSO was added in order to normalize the volume to 100 nL using D4 or D8 cassettes (Hewlett Packard, Palo Alto, CA, USA). Each plate contained positive and negative controls and was run in duplicate. After compounds were added, plates were processed as described above, per primary assay protocol. An average number of cells was calculated and plotted.

### 2.5. RNA Isolation, cDNA Synthesis, and Semi-Quantitative Real-Time (qRT) PCR

HEK-293, U-CH1, U-CH2, and HUVEC cells were grown, and RNA isolation was performed using the RNeasy Mini Kit (Qiagen, Germantown, MD, USA). DNAse treatment was performed, and first-strand cDNA synthesis was done for each RNA sample from 0.5 μg of total RNA using high-capacity reverse transcription reagents, as per manufacturer’s instructions (Thermo Fisher Scientific, Waltham, MA, USA).

### 2.6. qRT-PCR

Two μL of five-times-diluted cDNA was used for PCR using SyBr Green PCR Master Mix (Thermo Fisher Scientific, Waltham, MA, USA) in triplicate on a QuantStudio 7 thermal cycler (Thermo Fisher Scientific, Waltham, MA, USA) using *PSMB5-* and *PSMB8*-specific primers. *GAPDH* was used as a reference gene. Fold change was calculated with respect to HEK-293 using ∆∆Ct method. A list of primers is shown in Table 1.

### 2.7. CellTiter 96^®^ AQ_ueous_ One Solution Cell Proliferation Assay

Cell Titer 96^®^ AQ_ueous_ One Solution Cell Proliferation Assay was purchased from Promega (Cat. No. G5421) and the assay was performed as per the manufacturer’s guidelines. Briefly, 10,000 cells were plated in triplicate in a 96-well plate and incubated for 24 h. Cells were then treated with indicated concentrations of compounds and incubated for 72 h. Twenty-five microliters of CellTiter 96^®^ AQ_ueous_ One Solution Reagent (Promega, Madison, WI, USA) were added to the plate and incubated for 4 h. The absorbance of the wells in each plate was measured at 490 nm using a 96-well plate reader. Cells treated with DMSO were considered 100% proliferation, and graphs were plotted as percentage cell proliferation.

### 2.8. Western Blotting

Western blotting was performed as previously described [22]. Briefly, cells were lysed using RIPA buffer containing protease and phosphatase inhibitors. Cell lysates were cleared using centrifugation, and an equal protein of 20 μg was loaded onto an SDS polyacrylamide gradient gel (Bio-Rad, Hercules, CA, USA). Proteins were transferred onto a PVDF membrane and probed with PSMB5 and PSMB8-specific antibodies. β-Actin was used as a loading control. Species-specific secondary antibodies conjugated to Horse Radish Peroxidase (HRP) were incubated in order to detect primary antibodies after washes. Blots were developed using Luminata Forte HRP reagent with a Syngene (Fredrick, MD, USA) imaging system. Quantification of blots was performed using Syngene software version 1.8.6, and the data were normalized to the loading controls in order to calculate the fold change with respect to HEK-293 cells.

### 2.9. Purity Statement

The purity of the lead compounds is BNC105 96.6%, Palbociclib ≥ 98%, and Bortezomib ≥ 99%, as confirmed by vendors. PKS12265 > 95% and WZ1831 > 95% were confirmed by HPLC [23,24].

## 3. Results

**Development of a robust and reproducible primary assay for compound screening for chordoma:** In order to perform small molecule screening for chordoma, we utilized U-CH1 human chordoma cell line in a 384-well plate format. In the absence of a specific positive control, we used different concentrations of Doxorubicin as a positive control. We found a dose-dependent decrease in cell proliferation with 1 μM and 5 μM Doxorubicin (Figure 1A). Furthermore, we performed a five-point dose-response curve for Doxorubicin and 20% DMSO as a positive control for calculating the Z′-factor in duplicate plates (Figure 1B). We found a significant decrease in cell proliferation starting with concentrations as low as 0.5 μM doxorubicin (Figure 1B). Furthermore, we found that a lower concentration (0.1%) of DMSO did not affect cell proliferation, whereas a higher concentration of 5 μM DMSO did have an effect (Figure 1B). Based on these data, we selected Doxorubicin at 5 μM as a positive control for our screening. The standard Z′-factor for running a high-throughput assay is > 0.5, but depending on the assay complexity, a Z′-factor of lower than 0.5 is accepted for the assay [25,26]. Z′-factor values were calculated for each plate, and any plate with > 0.33 Z′-factor was included in the analysis, due to the large variations expected from an imaging-based screening. Plate histograms (Figure 1C,D) were used to evaluate the potential hits and to confirm if there are any patterns, such as the edge effect in each plate. We did not find edge effects or any significant patterns in any of the plates we ran, and negative and positive controls showed clear separation.

After confirming our assay’s robustness, we screened 4142 experimental wells containing 3730 unique compounds with diverse mechanisms of action and structures. The average number of cells from all the library plates screened was plotted. Our results showed clear decreases in cell count for the positive control (red spheres), and negative controls were segregated with most of the non-effective compounds (dark blue spheres). The experimental compounds are represented as X (light blue spheres), and those represented as Y (green spheres) are potential hits (Figure 2A). The correlation between each duplicate plate was measured, and a representative plate showed an R^2^ of 0.89, indicating that replicates show a good correlation (Figure 2B). Average Z-scores for compounds were plotted in order to identify the positive hits, and results indicated a decrease in Z-scores for potential hits, which indicates a decrease in the number of cells compared to the DMSO control (Figure 2C). The Z′-factors for each replicate plate were calculated and are shown in Appendix A. Our Z′-scores for all the plates showed higher values, confirming the robustness and reproducibility of all our library plates. Plates with a Z′-factor of 0.33 or higher were included in our analysis. As various libraries are curated at different concentrations depending on the activity, and some of the libraries are constructed for dose-dependent studies, we plotted the concentration-dependent plot for the potential hits, and the results suggested a dose-dependent decrease in average Z-scores (Figure 2D). Thus, these data confirm that the potential hits may be eligible for reconfirmation and dose-dependent studies following a cherry-pick. We performed primary screening for 4142 experimental wells with 3730 unique compounds and cherry-picked 50 potential hits for confirmation and dose-response studies (Figure 2E). Confirmed compounds with dose-dependent responses were selected as hits for further studies (Figure 2E).

**Cherry-pick and dose-response studies identify proteasomal inhibitors as hits for chordoma:** After analysis of the primary screening data, we cherry-picked 50 potential hits for confirmation and dose-response studies (Table 2). A list of compounds with their names, mechanisms of action, average Z-scores, and standard deviation are shown in Table 2. We found several potential hits from a diverse library of MedChem Express from different plates and six potential hits from the Cayman chemical library, as shown in Appendix A. We found 43 compounds reconfirmed for their activity at the same dose as the primary screen (Figure 3). The IC_50_ values for all the cherry-picked compounds are shown in Appendix A.

The dose-response curves were plotted for five compounds in each graph, as shown in Figure 3A–J. The IC_50_ values for the top 10 hits are shown in Table 3. The SMILES and structures of these top hits, and for Bortezomib in particular, are shown in Appendix A, respectively. In addition, we have included the list of 3730 unique compounds screened in Appendix A. Many compounds with sub-micromolar IC_50_ values may serve as potential therapeutic agents for chordoma after further studies.

Proteasomal inhibitors are potential therapeutic agents with specific chordoma cell-killing capabilities, either alone or in combination with anti-cancer drugs. From our primary screening, we found three proteasome inhibitors, ONX-0914, MG132, and Epoxomicin, as hits that demonstrated the reconfirmation (plotted separately in Figure 4A). Thus, we investigated specific proteasomal inhibitors including Bortezomib, which is approved for cancer therapy. Moreover, we investigated the proteasomal inhibitor Bortezomib in combination with other classes of anti-cancer drugs in order to investigate the synergistic effects, based on previous publications, with Bortezomib on cancer cell killing, either alone or in a combination of microtubule-binding drug and CDK4/6 inhibitors [27,28]. We selected BNC105, a microtubule-binding drug used as a vascular targeting agent for cancer, and Palbociclib, a CDK4/6 inhibitor approved for cancer therapy, and compared the cell-killing efficiency of these substances to that of Bortezomib, either alone or in combination. These compounds show dose-dependent decreases in cell proliferation at higher doses, which were more prominent at lower doses in combination with Bortezomib (Figure 4B,C). In order to investigate the specificity of proteasomal inhibitor Bortezomib with anti-cancer drugs, we utilized the non-cancerous cell line HEK-293 and compared it with U-CH1 and U-CH2 cells. Our data confirmed increased killing by proteasomal inhibitors, either alone or in synergy in U-CH1 and U-CH2 cells, compared to HEK-293 cells (Figure 4D–F). In addition, our data showed increased killing by combination therapy compared to individually treated U-CH2 cells (Figure 4G). Furthermore, we tested the efficacy of PKS21265 [23] and WZ1831 (β5i- and β5c-selective inhibitors, respectively), Bortezomib (BTZ, a co-inhibitor of both β1 and β5), and Carfilzomib (CFZ, an inhibitor of β5) in U-CH1 cells and found that the IC_50_ values of BTZ are lower than those of CFZ and selective inhibitors of β5i and β5c (Figure 4H). Thus, these data show a reduction in the proliferation of chordoma cells with a promise of therapeutic targeting of chordoma cells with proteasomal inhibitors, alone or in combination with other chemotherapeutic agents. Our data also show the selective killing of chordoma cells compared to non-cancerous cells, indicating the specificity towards chordoma cells. We calculated the IC_50_ values for different cells and found that Bortezomib decreased IC_50_ in chordoma cells compared to non-cancerous cells (Table 4). Further investigation of the synergistic effect of Bortezomib with anti-cancer drugs showed a decrease in IC_50_ values in U-cH2 cells after a combination of the drugs (Table 4). In summary, we have identified and reconfirmed at least 10 top potential compounds which show promising results in reducing chordoma cell proliferation after screening 3730 unique compounds from diverse libraries.

**Proteasomal subunits PSMB5 and PSMB8 are highly upregulated in chordoma cell lines U-CH1 and U-CH2:** As we found, proteasomal inhibitors as one class of the hits from primary screening; next, we asked whether proteasomal subunits are increased in chordoma. We performed the gene expression analysis of proteasomal subunits *PSMB5* and *PSMB8* by qRT-PCR in U-CH1 and U-CH2 cells, as compared to non-cancerous cell lines HEK-293 and HUVEC. Our data show a significant increased expression for both *PSMB5* and *PSMB8* subunits in these two chordoma cells compared to both HEK-293 and HUVEC cells (Figure 5A,B). Additionally, our Western blotting results showed increased expression of PSMB8 in U-CH1 and U-CH2 cells as compared to HEK-293 cells, and increased expression of PSMB5 only in U-CH2 cells as compared to HEK-293 cells (Figure 5C). Thus, these data confirm upregulated gene and protein expression of proteasomal subunits *PSMB5* and *PSMB8* in these two chordoma cell lines, as compared to normal cells.

## 4. Discussion

Chordoma is a rare bone cancer, often only detected in advanced stages. Due to the lack of approved therapies for chordoma, there is an immediate need for therapeutic agents that can reduce its growth. Previously, Xia et al. conducted a small molecule screening with an FDA-approved compound and found Bortezomib, a proteasomal inhibitor, as a hit [12]. Small molecule screening of chordoma using U-CH1 and U-CH2 cells showed significant sensitivity for NFκB inhibitory pathways and tested sunitinib and Bortezomib in the chordoma xenograft model [29]. In a recent study, Sharifnia et al. found THZ1, a brachyury inhibitor, effective against chordoma [13]. Using 133 clinically approved compound screenings, Scheipl et al. found improved efficacy of EGFR inhibitors in combination with HDAC and Topoisomerase inhibitors [30]. In accordance with the above studies, we found three proteasomal inhibitors that act as potential hits for chordoma. For the first time, we performed the gene expression analysis for proteasomal subunits *PSMB5* and *PSMB8,* and found that these two subunits are increased in chordoma. These findings provide a molecular basis for targeting the proteasomal complex using these proteasomal inhibitors. In addition to those available proteasomal inhibitors, we found that chemotherapeutic agents BNC105 and Palbociclib alone, or combined with Bortezomib, decreased chordoma cell proliferation. Thus, Bortezomib alone, or in combination with BNC105 or Palbociclib, shows promise for therapeutic efficacy at lower doses, which may reduce the toxicity to normal cells, making these compounds safe and specific for in vivo studies.

The ubiquitin-proteasome system (UPS) is the most important pathway by which cytosolic protein degradation occurs. Proteins destined for proteasomal degradation are tagged with poly-ubiquitin chains via a cascade of ubiquitination reactions. Proteasome 20S is the core particle that hydrolyzes ubiquitinated proteins. There are two copies of three protease-like proteolytic active subunits in each core particle: caspase-like β1, tryptic-like β2, and chymotryptic β5. In humans, there are two major proteasomes: constitutive proteasome in all cells, and immunoproteasome in immune cells. Non-immune cells can also express immunoproteasome at inflammation sites or through stimulation with proinflammatory cytokines. Human Proteasome is a validated target for hematologic malignancies, with three drugs approved by FDA: bortezomib, ixazomib, and carfilzomib [31,32]. All three are pan-β5 inhibitors, as they equally inhibit β5c of constitutive proteasome and β5i of the immunoproteasome. A specific inhibitor of β5i is a novel target for autoimmune diseases validated in many animal models [33].

Proteasome 20S subunit beta 5 (PSMB5) is increased in various cancers [34,35,36,37], and its mutation leads to resistance to proteasomal inhibitors [34,38]. Similar to PSMB5, Proteasome 20S subunit beta 8 (PSMB8) overexpression correlates with the progression of gastric cancer, its inhibition in glioblastoma increases apoptosis and decreases angiogenesis [39,40], and its mutation leads to inflammation and lipodystrophy [41]. Proteasomal inhibitors have been in clinical trials for hematological malignancies, including acute lymphoblastic leukemia [42], acute myeloid leukemia [43,44,45,46], and myelodysplastic syndrome [45], but the results were not promising [42,45,46,47]. The bioavailability of Bortezomib in solid tumors was limited, with a possibility of the development of resistance. A higher dose was applied, but led to enhanced toxicity [48]. Bortezomib was approved for early relapse [32]. Thus, novel proteasome inhibitors, such as carfilzomib or ixazomib, are in clinical trials with lower doses for solid tumors (NCT00531284, NCT01949545).

The primary targets involved in chordoma are receptor tyrosine kinases (RTKs) and their downstream signaling, including the PI3K/AKT/mTOR [49]. Reports studying the efficacy of several RTK inhibitors found the following inhibitors—imatinib, sunitinib, vandetanib, gefitinib, PI3K/mTOR inhibitors, BEZ-235, rapamycin, and p38 MAP kinase inhibitors (SB 202190 and SB 203580)—either inactive or less effective on chordoma cells [49]. Only erlotinib showed a 27% decrease in the proliferation [12]. Similar to other carcinomas, the activity of various kinases is increased in chordoma, including vascular endothelial growth factor receptor (VEGFR), EGFR, and PDGFR [50]. These kinase inhibitors have been approved for clinical cancer treatments or are in advanced clinical trials. Sunitinib is in phase I–II clinical trials for treating AML and is approved for treating renal cell cancer. Imatinib is given to patients with chronic myeloid leukemia (CML) and gastrointestinal stromal tumors (GISTs). BEZ-235 shows promising antitumor activity in vivo [51] and is currently in phase I and II clinical trials for solid tumors. These results suggest that chordoma cell growth is not dependent on these pathways and, thus, the need to identify the crucial signaling pathways essential for chordoma cell survival. Recently, we and others have found novel highly selective proteasomal inhibitors have been synthesized, which have the potential to be tested for chordoma and other cancers with increased proteasomal subunit b5c [24,52]. Thus, our novel compounds, including proteasomal inhibitors, offer an innovative tool for studying the synergistic effects of compounds that show low efficiency toward chordoma growth inhibition (Figure 6).

## 5. Conclusions

We have identified small molecules, including proteasomal inhibitors, which show promise for cell growth inhibition in human chordoma cell lines. Thus, our study provides significant results to advance these small molecular and proteasomal inhibitors for preclinical studies involving small animals. Among all hits, we identified and confirmed three proteasomal inhibitor hits that inhibited U-CH1 cell proliferation. Furthermore, we found that proteasomal subunits proteasome 20S subunit beta 5 (PSMB5) and proteasome 20S subunit beta 8 (PSMB8) upregulated in two human chordoma cell lines, U-CH1 and U-CH2. Bortezomib alone, or in combination with Palbociclib or BNC105, decreased the proliferation of chordoma cell line U-CH2. The synergistic effects of growth inhibition indicate a potential combination therapy that could be tested in chordoma patients.

## Figures and Tables

**Figure 1 pharmaceutics-15-01274-f001:**
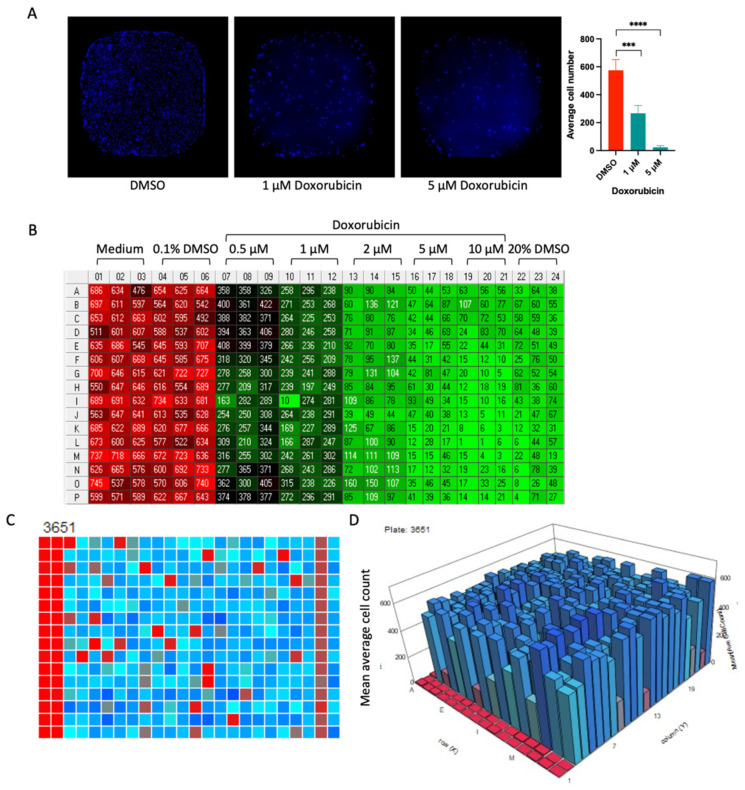
**Development of a primary assay for screening the compound.** (**A**) Representative images of cells following doxorubicin treatment. Quantitation of cell number shows a significant decrease in proliferation with doxorubicin. *** and **** represent *p* < 0.001 and 0.0001, respectively. (**B**) A representative plate showing cell numbers as counted by Hoechst staining for Z-prime calculation. Medium and 0.1% DMSO as negative control, and different concentrations of doxorubicin and 20% DMSO as positive controls. (**C**) Mean average cell numbers plotted in a plate format. Plate showing positive controls (5 μM doxorubicin, 1st^,^ and 2nd columns, 20% DMSO, 23rd column), negative controls (0.1% DMSO) 24th column, and experimental compounds 3–22 columns. Color range Red 780 and green 320. (**D**) Mean of average cell counts was plotted to identify edge effects.

**Figure 2 pharmaceutics-15-01274-f002:**
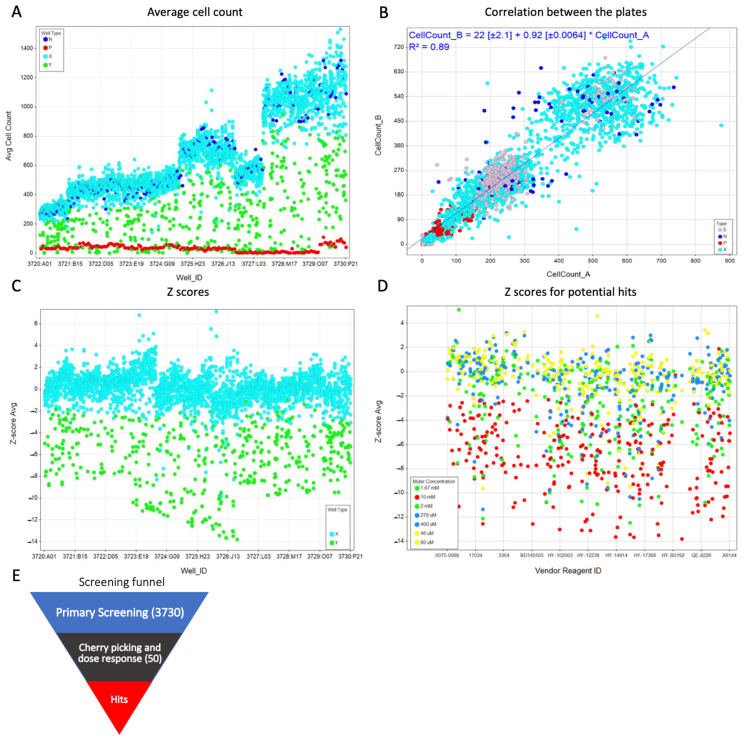
**Small molecule screening identifies potential hits.** (**A**) Average cell number was plotted for all the plates screened. Red represents positive controls, dark blue represents negative control, light blue represents non-hit compounds, and green represents potential hits. (**B**) Representative plot for the correlation of replicate plates shows a high correlation between replicates. Red represents positive controls, dark blue represents negative control, light blue represents non-hit compounds, and gray represents potential hits. (**C**) Average Z-scores for each compound were calculated and plotted. Light blue represents non-hit compounds, and green represents potential hits. (**D**) Z-scores for various concentrations of selected potential hits were calculated. (**E**) Summary of primary screening showing a number of compounds for cherry-picking and confirmation of potential hits.

**Figure 3 pharmaceutics-15-01274-f003:**
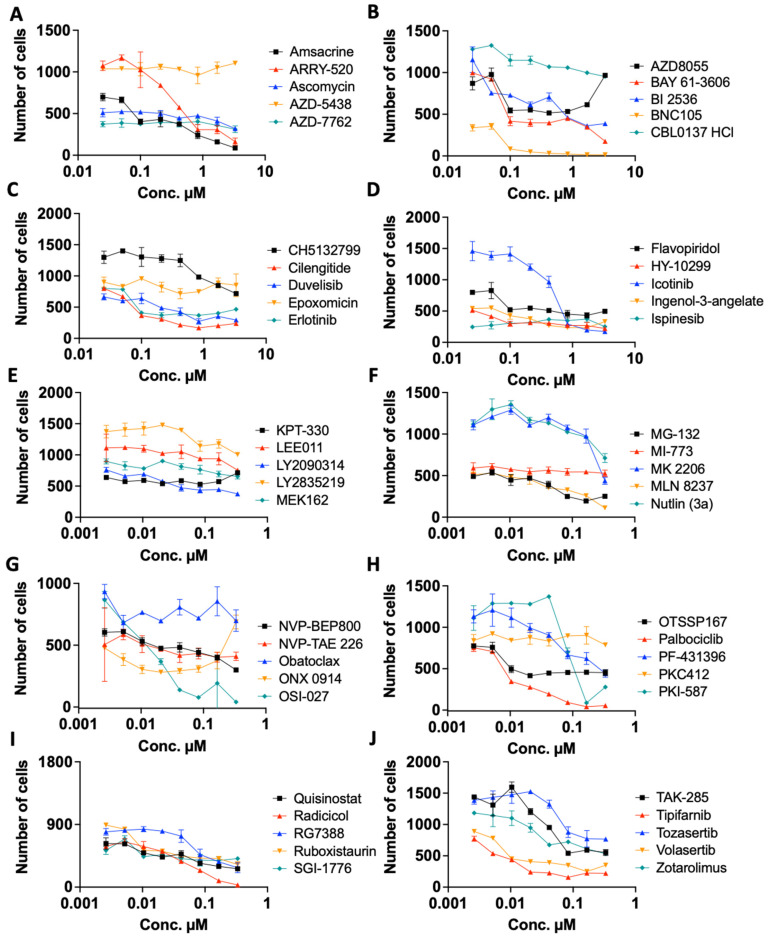
**Reconfirmation of potential hits from primary assays and dose dependency studies identifies proteasomal inhibitors**. (**A**–**J**) Dose-response curves for cherry-picked compounds from primary screening done in duplicate. (**F**) Dose dependency curves for three proteasomal inhibitors showing reconfirmation of these three inhibitors. (**G**,**H**) Dose dependency curves for BNC105 and bortezomib or Palbociclib alone or in combination showing low-dose synergistic cell-killing activity of chordoma cells.

**Figure 4 pharmaceutics-15-01274-f004:**
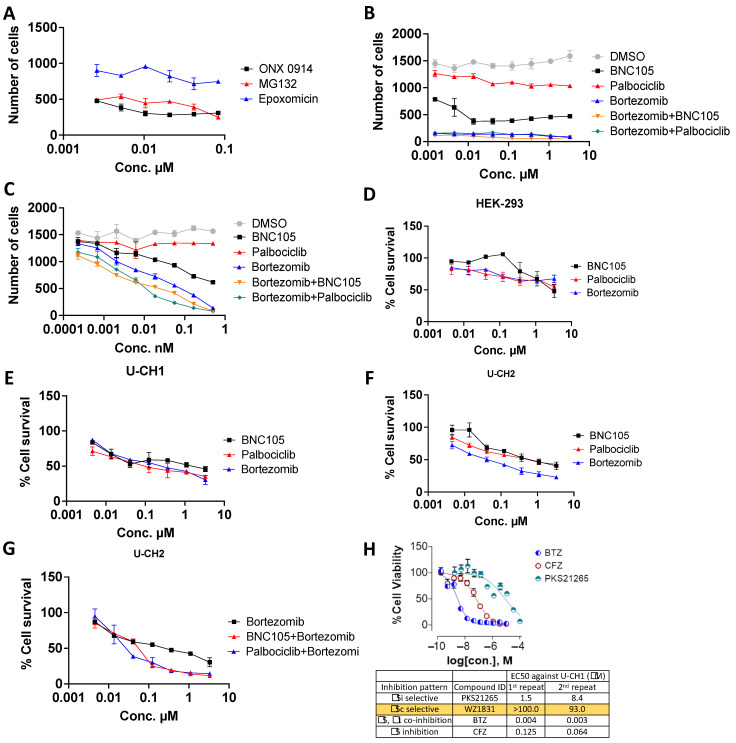
**Primary assays identify proteasomal inhibitors as a specific therapeutic agent, alone or in combination with other anti-cancer drugs**. (**A**–**C**) Nuclear count. (**A**) Dose dependency curves for three proteasomal inhibitors showing reconfirmation of these three inhibitors. (**B**) High-dose and (**C**) Low-dose dependency curves for Bortezomib and BNC105 or Palbociclib alone or in combination, showing low-dose synergistic cell-killing activity of chordoma cells. Dose dependency study using proliferation assays (**D**) HEK-293 cells, (**E**) U-CH1 and (**F**) U-CH2 cells alone or (**G**) in combination. (**H**) Percentage cell survival evaluated for BTZ, CFZ, and PKS21265 and the IC_50_ values for β5i- and β5c-selective and co-inhibition in U-CH1 cells.

**Figure 5 pharmaceutics-15-01274-f005:**
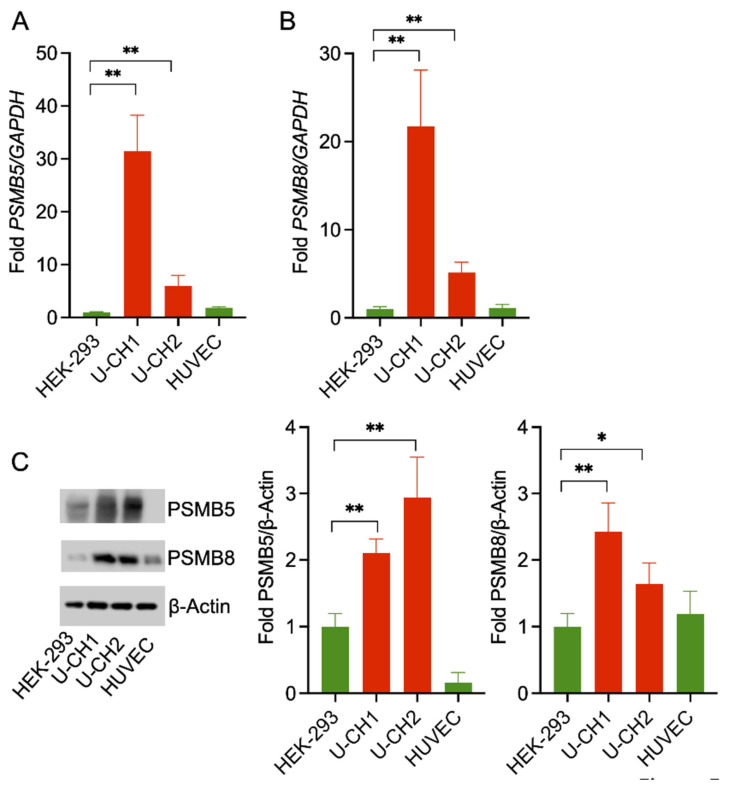
**Proteasomal subunits *PSMB5* and *PSMB8* are increased in chordoma cells U-CH1 and U-CH2.** Gene expression analysis by RT PCR represented as a fold change of (**A**) *PSMB5* and (**B**) *PSMB8,* showing increased expression of PSMB5 and PSMB8 genes in U-CH1 and UCH1 cells as compared to HEK-293 and HUVEC cells. ** represents *p* < 0.01. (**C**) Western blotting for PSMB5 and PSMB8 in HEK-293, U-CH1, U-CH2, and HUVEC cells showed increased protein expression of these two proteins in chordoma cells. β-Actin served as a loading control. Quantitation of PSMB5 and PSMB8 protein expression normalized to β-Actin. * and ** represents *p* ≤ 0.05 and *p* ≤ 0.01 respectively.

**Figure 6 pharmaceutics-15-01274-f006:**
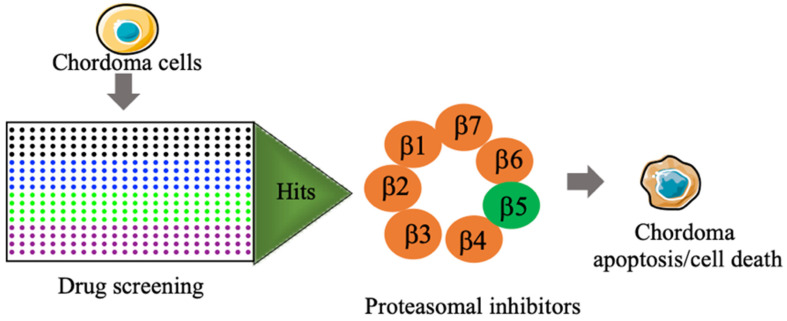
**Schematic diagram showing small molecule inhibitor/proteasomal inhibitors for reducing chordoma cell proliferation**.

**Table 1 pharmaceutics-15-01274-t001:** List of primers used for RT-PCR.

List of Primers Used for RT-PCR	
PSMB5 F	5′-AGGAACGCATCTCTGTAGCAG-3′
PSMB5 R	5′-AGGGCCTCTCTTATCCCAGC-3′
PSMB8 F	5′-CCTTACCTGCTTGGCACCATGT-3′
PSMB8 R	5′-TTGGAGGCTGCCGACACTGAAA-3′
GAPDH F	5′-ACAACTTTGGTATCGTGGAAGG-3′
GAPDH R	5′-GCCATCACGCCACAGTTTC-3′

**Table 2 pharmaceutics-15-01274-t002:** **List of cherry-picked compounds.** The top 50 compounds were selected as positive hits after the primary screen. Average Z-score and standard deviation between the two replicate plates were considered as criteria for selecting these potential hits. The table shows compounds with different mechanisms of action were qualified for cherry-picking, including the proteasomal inhibitors.

Compound Name	Reagent ID	Average Z-Score	Z-Score (SD)	Function	Vendor
Amsacrine	HY-13551	−8.79	0.141	A potent intercalating antineoplastic agent. It is effective in the treatment of acute leukemias and malignant lymphomas but has poor activity in the treatment of solid tumors.	Medchem Express
ARRY-520	HY-15187	−7.33	0.332	Filanesib (ARRY-520) is a synthetic kinesin spindle protein (KSP) inhibitor	Medchem Express
Ascomycin	11309	−7.56	1.782	Ascomycin inhibits the production of Th1 (interferon- and IL-2) and Th2 (IL-4 and IL-10) cytokines.	Cayman Chemical
AZD-5438	HY-10012	−7.19	0.148	AZD-5438 is a potent inhibitor of CDK1/2/9	Medchem Express
AZD-7762	HY-10992	−9.02	0.403	AZD-7762 is a potent ATP-competitive checkpoint kinase (Chk) inhibitor	Medchem Express
AZD8055	SS-4787	−7.15	0.919	AZD-8055 is a novel ATP-competitive inhibitor of mTOR kinase. AZD-8055 inhibits both mTORC1 and mTORC2.	Key Organics
BAY 61-3606	HY-14985	−9.85	4.172	BAY-61-3606 is a potent and selective inhibitor of Syk kinase	Medchem Express
BI 2536	HY-50698	−7.88	1.902	Bl2356 is a notable dual PLK1 and BRD4 inhibitor. BI-2536 suppresses IFNB (encoding IFN-β) gene transcription.	Medchem Express
BNC105	HY-16114	−6.33	0.346	BNC105 is a tubulin polymerization inhibitor with potent antiproliferative and tumor vascular disrupting properties.	Medchem Express
CBL0137 HCl	HY-18935A	−6.77	0.792	CBL0137(CBL-0137) activates p53 and inhibits NF-kB	Medchem Express
CH5132799	HY-15466	−5.87	0.354	CH5132799 is a selective class I PI3K inhibitor	Medchem Express
Cilengitide	HY-16141	−10.05	4.879	Cilengitide is a cyclic RGD-containing peptide that binds cancer cells expressing high concentrations of αVβ3 and αVβ5 integrins.	Medchem Express
Duvelisib	HY-17044	−3.98	0.219	Duvelisib is a Phosphoinositide 3-kinase inhibitor, specifically of the delta and gamma isoforms of PI3K	Medchem Express
Epoxomicin	HY-13821	−9.65	0.134	Epoxomicin (BU 4061T) is a naturally occurring selective proteasome inhibitor with anti-inflammatory activity. Epoxomicin covalently binds to the LMP7, X, MECL1, and Z catalytic subunits of the proteasome.	Medchem Express
Erlotinib	T0373	−8.25	0.354	Erlotinib is an epidermal growth factor receptor inhibitor (EGFR inhibitor).	Target Molecule
Flavopiridol	HY-10005	−9.70	3.960	Flavopiridol (Alvocidib) competes with ATP to inhibit CDKs including CDK1, CDK2, CDK4, CDK6, and CDK9	Medchem Express
GSK-923295	HY-10299	−7.66	0.219	Allosteric inhibitor of centromere-associated protein-E (CENP-E) kinesin motor ATPase activity	Medchem Express
Icotinib	HY-15164	−5.31	0.686	Icotinib Hydrochloride (BPI-2009) is a potent and specific EGFR inhibitor	Medchem Express
Ingenol-3-angelate	16207	−3.81	1.072	Ingenol 3-Angelate is a protein kinase C activator	Cayman Chemical
Ispinesib	HY-50759	−13.50	4.243	Ispinesib selectively inhibits the mitotic motor protein, kinesin spindle protein (KSP), resulting in inhibition of mitotic spindle assembly, induction of cell cycle arrest during the mitotic phase, and cell death in tumor cells that are actively dividing. Because	Medchem Express
KPT-330	S7252	−10.72	0.205	KPT-330, analog of KPT-185, is a selective inhibitor of CRM1. CRM1 is a nuclear export receptor involved in the active transport of transcription factors, cell-cycle regulators, tumor suppressors and RNA molecules.	Selleck Chemicals
LEE011	HY-15777	−3.76	0.035	Ribociclib (LEE01) is a highly specific CDK4/6 inhibitor	Medchem Express
LY2090314	HY-16294	−5.87	1.478	LY2090314 is a potent inhibitor of glycogen synthase kinase-3	Medchem Express
LY2835219	HY-16297	−7.98	2.001	Cyclophosphamide is a synthetic alkylating agent chemically related to the nitrogen mustards with antineoplastic activity, an immunosuppressant.	Medchem Express
MEK162	HY-15202	−4.32	0.792	Binimetinib (MEK162) is an oral and selective MEK1/2 inhibitor.	Medchem Express
MG-132	HY-13259	−10.90	0.417	MG132 is a potent cell-permeable proteasome and calpain inhibitor	Medchem Express
MI-773 (SAR405838)	S7649	−8.13	0.488	MI-773 is an Inhibitor of the MDM2-p53 interaction	Selleck Chemicals
MK 2206	HY-10358	−6.34	0.141	MK-2206 dihydrochloride (MK-2206 (2HCl)) is an allosteric AKT inhibitor	Medchem Express
MLN 8237	2003	−6.69	0.424	Alisertib (MLN 8237) induces apoptosis and autophagy through targeting the AKT/mTOR/AMPK/p38 pathway in leukemic cells. Antitumor activity.	Axon Medchem
Nutlin (3a)	HY-10029	−7.81	0.021	Nutlin 3a is an active enantiomer of Nutlin-3, acts as a murine double minute (MDM2) inhibitor that inhibits MDM2-p53 interactions and stabilizes the p53 protein, and thereby induces cell cycle arrest and apoptosis.	Medchem Express
NVP-BEP800	18383	−5.33	1.287	Ingenol-3-angelate causes inflammation due, at least in part, to activation of PKC, leading to antibody-dependent cellular cytotoxicity	Cayman Chemical
NVP-TAE 226	HY-13203	−7.35	1.407	NVP-TAE 226 (TAE226) is a potent and ATP-competitive dual FAK and IGF-1R	Medchem Express
Obatoclax	HY-10969	−13.05	4.313	Obatoclax is an inhibitor of the Bcl-2 family of proteins.	Medchem Express
ONX 0914	16271	−8.81	1.882	ONX is a selective inhibitor of the β5i (LMP7) subunit of the immunoproteasome	Cayman Chemical
OSI-027	HY-10423	−5.39	0.085	OSI-027 is an ATP-competitive mTOR kinase activity inhibitor. OSI-027 targets both mTORC1 and mTORC2	Medchem Express
OTSSP167 (hydrochloride)	HY-15512A	−13.40	4.384	OTSSP167, also known as OTS167, is an orally available inhibitor of maternal embryonic leucine zipper kinase (MELK) with potential antineoplastic activity.	Medchem Express
palbociclib	THR0011	−5.70	0.849	Selective inhibitor of the cyclin-dependent kinases CDK4 and CDK6	Pharmablock
PF-431396	HY-10460	−7.41	0.021	PF-431396 is a potent and selective focal adhesion kinase (FAK) and proline-rich tyrosine kinase 2 (PYK2) inhibitor	Medchem Express
PKC412	HY-10230	−7.05	1.704	Midostaurin (PKC412; CGP 41251) is a multi-targeted protein kinase inhibitor which inhibits PKCα/β/γ, Syk, Flk-1, Akt, PKA, c-Kit, c-Src, PDFRβ and VEGFR1/2	Medchem Express
PKI-587	HY-10681	−8.36	0.035	Gedatolisib, also known as PKI-587 and PF-05212384, is an agent targeting the phosphatidylinositol 3 kinase (PI3K) and mammalian target of rapamycin (mTOR) in the PI3K/mTOR signaling pathway, with potential antineoplastic activity	Medchem Express
Quisinostat	HY-15433	−8.58	2.107	Quisinostat is a "second generation" histone deacetylase inhibitor with antineoplastic activity.It is highly potent against class I and II HDACs	Medchem Express
Radicicol	BIR0140	−7.70	0.495	Radicicol inhibits the activities of Hsp90, Topo VI and PDK3 by blocking ATP binding to them.	Apollo Scientific
RG7388	HY-15676	−9.25	0.141	Idasanutlin (RG7388) is a potent and selective MDM2 antagonist	Medchem Express
Ruboxistaurin	HY-10195B	−13.00	4.808	LY333531 is a potent inhibitor of protein kinase Cβ	Medchem Express
SGI-1776	HY-13287	−8.44	1.768	SGI-1776 is an inhibitor of Pim kinases	Medchem Express
TAK-285	HY-15196	−5.74	0.955	TAK-285 is a potent, selective, ATP-competitive and orally active HER2 and EGFR(HER1) inhibitor	Medchem Express
Tipifarnib	HY-10502	−9.51	0.120	Tipifarnib, is a farnesyltransferase inhibitor. It inhibits the Ras kinase in a post translational modification step before the kinase pathway becomes hyperactive.	Medchem Express
Tozasertib	HY-10161	−6.29	0.339	Tozasertib (VX 680; MK-0457) is an inhibitor of Aurora A/B/C kinases	Medchem Express
Volasertib	HY-12137	−12.15	0.071	BI6727 (Volasertib) is a selective inhibitor of Plk1, Plk2, and Plk3	Vitas M Labs
Zotarolimus	HY-12424	−4.31	0.587	Zotarolimus is mechanistically similar to sirolimus in having high-affinity binding to the immunophilin FKBP12 and comparable potency for inhibiting in vitro proliferation of both human and rat T cells.	Medchem Express

**Table 3 pharmaceutics-15-01274-t003:** List of IC_50_ values for top 10 hits.

Compound Name	IC_50_ (μM)
ARRY-520	1.086
BAY 61-3606 (dihydrochloride)	1.229
BNC105	0.9265
Bortezomib	2.31
Icotinib (Hydrochloride)	1.129
MG-132	0.03263
MLN 8237	0.02419
OSI-027	0.9978
Palbociclib	0.1067
Tipifarnib	0.9954

**Table 4 pharmaceutics-15-01274-t004:** List of IC_50_ values for Bortezomib and anti-cancer drugs in HEK-293 and chordoma cells, individually or in combination.

Compound Name	IC_50_ (μM)
**HEK-293**	
BNC105	0.8489
Bortezomib	0.8969
Palbociclib	0.9903
**U-CH1**	
BNC105	0.9539
Bortezomib	0.6041
Palbociclib	0.5906
**U-CH2**	
BNC105	0.2939
Bortezomib	0.5761
Palbociclib	0.7048
**U-CH2**	
Bortezomib	0.6041
BNC105+Bortezomib	0.3805
Palbociclib+Bortezomib	0.4477

## Data Availability

The data used in this manuscript are available for the research community upon reasonable request from the corresponding author.

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
