# Peer review of "High-Throughput/High Content Imaging Screen Identifies Novel Small Molecule Inhibitors and Immunoproteasomes as Therapeutic Targets for Chordoma"

_pharmaceutics, 2023, doi:10.3390/pharmaceutics15041274_

Round 1

Reviewer 1 Report (Previous Reviewer 2)

Dear Authors,

You presented a quite nice research, however it is not complete.

You pointed, that you tested 4000 compounds and finally selected 50 for the experiment.

Why you did not attached an excel table with all those compounds as supplementary information?

It will help others with the research.

In the manuscript is it hard to follow the compound role and properties.

Could you implement compound numbers? It will allow to easy follow it (see table 3, 4 and fig 3), or at least make it in alphabetic order.

Minor revision:

Show all tested compounds in supplementary data.

Make order with compounds present in the paper. (cas numbers, would be nice).

Author Response

Reviewer 1:

Comment: Dear Authors,

You presented a quite nice research, however it is not complete. You pointed, that you tested 4000 compounds and finally selected 50 for the experiment. Why you did not attached an excel table with all those compounds as supplementary information? It will help others with the research.

Response to the comment: We appreciate the reviewer’s comments towards improving our manuscript. We have highlighted all the revised texts in blue color. It is a good suggestion to include the list of all the compounds we screened. We have added them in Supplementary table 1 in the revised manuscript.

Comment: In the manuscript is it hard to follow the compound role and properties. Could you implement compound numbers? It will allow to easy follow it (see table 3, 4 and fig 3), or at least make it in alphabetic order.

Response to the comment: Thank you for these suggestions, and our apologies for not making it in alphabetical order. Per the reviewer’s suggestion, we have made the list of compounds in alphabetic order for each compound in the revised manuscript. After removing the overlapping compounds from the 4142 experimental well screened, we found 3730 unique compounds. We have updated this information in our revised manuscript.

Minor revision:

Comment: Show all tested compounds in supplementary data.

Response to the comment: We have added the list of all the compounds tested in the Supplementary Table 1 of the revised manuscript.

Comment: Make order with compounds present in the paper. (cas numbers, would be nice).
Response to the comment: We have added the order of the compounds by arranging each compound alphabetically.

Reviewer 2 Report (New Reviewer)

This paper is a study of screening small molecules to find treatment targets for chordoma.

Chordoma is a disease that needs to be solved, and finding a target molecule for treatment is important. I read this paper with interest. However, it is questionable whether it is an appropriate level for a Pharmaceutics journal (IF=6.525). The following should be corrected or resolved.

1-1. Please consider suggesting how many and which targets were specifically presented in the abstract. It would also be meaningful to directly present the top 3 of the compounds.

1-2. If my understanding is correct, the compounds in Table 2 out of 4142 were presented. Please consider presenting accurate figures.

2. Does T gene mean TBXT gene? Please consider presenting with the TBXT gene.

3. A subtitle must be presented in the method part. Please list a total of 9 subtitles, 2.1. 2.2. 2.3.

4. The picture on 14p is incorrectly inserted. Please suggest an appropriate layout.

5. (Most important) For the compounds in Table 2, how many compounds should be presented. Please briefly present the legend as to why it was presented. The selected evidence must be presented (statistical significance, etc.), and please present it by adding one column.

6. In the case of Table 3 and Table 4, simple scientific information is presented. Please consider merging.

Author Response

Reviewer 2

This paper studies screening small molecules to find treatment targets for chordoma. Chordoma is a disease that needs to be solved, and finding a target molecule for treatment is important. I read this paper with interest. However, it is questionable whether it is an appropriate level for a Pharmaceutics journal (IF=6.525). The following should be corrected or resolved.

Comment: 1-1. Please consider suggesting how many and which targets were specifically presented in the abstract. It would also be meaningful to directly present the top 3 of the compounds.

Response to the comment: We thank the reviewer for their positive comment to enhance the quality of our manuscript. We appreciate the reviewer’s scientific input and valuable time. We have highlighted all the revised texts in blue color. We have added the number of targets in the abstract and added the top three compounds in the abstract.

Comment: 1-2. If my understanding is correct, the compounds in Table 2 out of 4142 were presented. Please consider presenting accurate figures.

Response to the comment: Yes, the reviewer is correct, we have presented the cherry-picked compound from the primary screening of 4142 compounds in Table 2. These 50 compounds are the top potential hits based on their Z score and standard deviation between the plates.

Comment: 2. Does T gene mean TBXT gene? Please consider presenting with the TBXT gene.

Response to the comment: Yes, the reviewer is correct, the T gene means TBXT gene. We have replaced the T gene name with the TBXT gene.

Comment: 3. A subtitle must be presented in the method part. Please list a total of 9 subtitles, 2.1. 2.2. 2.3.

Response to the comment: We have added the subtitles as suggested by the reviewer.

Comment: 4. The picture on 14p is incorrectly inserted. Please suggest an appropriate layout.

Response to the comment: We have corrected the orientation of the picture and added the legends to it.

Comment: 5. (Most important) For the compounds in Table 2, how many compounds should be presented. Please briefly present the legend as to why it was presented. The selected evidence must be presented (statistical significance, etc.), and please present it by adding one column.

Response to the comment: Table two presents the cherry-picked compounds, their vendor ID, and their known mechanism of action. We have added the criteria for selecting them for cherry-picking. Briefly, primary screening data were analyzed to calculate the Z score for each compound based on its ability to reduce cell proliferation. Then the standard deviation between the replicate plates A and B was considered for the scoring. Based on these criteria top 50 compounds with a significant decrease in cell proliferation and similar capability to reduce the proliferation in both the replication were selected for cherry picking. We have included these in the revised manuscript.

Comment: 6. In the case of Table 3 and Table 4, simple scientific information is presented. Please consider merging.

Response to the comment: Table 3 and Table 4 are done separately to make two different claims. Table 3 data represents the dose and reconfirmation studies from the original compound library, whereas Table 4 data is from the compounds we procured from the vendor. Moreover, Table 1 only uses 1 cell line, but in Table 4, we used three cell lines. Finally, in Table 4, we studied the synergistic effect of compounds to inhibit the antiproliferative effect on chordoma cells. Thus, combining these two tables may confuse the readers when we explain the results in our manuscript. We have alphabetically made the list of compounds so readers can follow it easily.

Round 2

Reviewer 2 Report (New Reviewer)

Dear Authors,

The Manuscript has been fine-tuned based on the reviewer's comments.

It is expected to be published once some minor issues are resolved (forms, figure resolution, etc.).

Author Response

Thank you for your kind review. We have made changes suggested by the academic editor and incorporated your suggestions.

This manuscript is a resubmission of an earlier submission. The following is a list of the peer review reports and author responses from that submission.

Round 1

Reviewer 1 Report

In their manuscript submitted to “Pharmaceutics”, Ajay et al. describe the results of a small molecule screen with the U-CH1 chordoma cell line to identify chemicals that inhibit the proliferation of these cells and can serve as potential therapeutic agents for the treatment of chordoma patients. They start with a single-concentration screen comprising 4241 molecules and then perform dose-dependency studies with 50 promising candidates. They found that proteasome inhibitors in particular reduced the proliferation of U-CH1 cells. Furthermore, they test combinations of the proteasome inhibitor Bortezomib with BNC105 or Palbociclib at low doses for their potential to inhibit U-CH1 cell proliferation. Finally, they claim that the proteasomal subunits PSMB5 and PSMB8 are overexpressed in U-CH1 and U-CH2 cells and suggest that this finding provides a molecular basis for targeting the proteasomal complex in chordoma patients.

Given that chordomas are a very rare form of cancer and that clinical studies comprising larger patient cohorts are difficult to perform, analyzing the effects of drugs with cell-based assays is important for the identification of potential therapeutics. However, for several reasons mentioned below, I am not enthusiastic about this manuscript and cannot recommend publication in its present form. I suggest that the authors improve their experimental design, conduct additional experiments, carefully revise the manuscript, and then submit it to a more specialized journal.

1.     As the authors mention, a comparable small molecule screen with 2800 drugs has been published already in 2013 (Xia et al., PMID: 23792643). In this screen, 15 different concentrations of all drugs were tested for their potential to inhibit the proliferation of U-CH1 and U-CH2 cells and a non-neoplastic control cell line. 35 of these compounds were further tested with primary cell lines derived from chordoma patients. In their screen, Xia et al. identify the proteasome inhibitor Bortezomib as the most potent agent to inhibit chordoma cell proliferation. Proteasome inhibitors were shown in additional publications to be efficient in inhibiting the proliferation of chordoma cells and were analyzed in chordoma xenograft models (e.g. PMID: 34550531, PMID: 24223206; both publications not cited by Ajay et al.). All these studies found that proteasome inhibitors can strongly inhibit the growth of chordoma cells. Given that Ajay et al. mainly focus on analyzing a single chordoma cell line (U-CH1) and only confirm the results of several much more detailed studies, the submitted manuscript does not provide a substantial gain in knowledge on treatment options for chordoma patients.

2.     The authors tested the compounds exclusively with the U-CH1 chordoma cell line and did not treat non-malignant cells in parallel. Such control experiments would have been particularly important since proteasome inhibitors were shown to reduce the viability of a broad set of cell types due to the accumulation of toxic ubiquitinylated proteins. Thus, the results submitted by Ajay et al. do not provide evidence for selective cytotoxicity against chordoma cells which further reduces the value of the screen.

3.     In Figs. 3A-3E, the authors show dose-response curves for 10 compounds per figure. Due to the style of the diagrams, it is impossible to identify the particular curves for most of the compounds.

4.     In Figs. 3G and 3H, the authors show results about the dose-dependent inhibition of U-CH1 cell proliferation with Bortezomib, BNC105, and Palbociclib as well as combinations of the compounds. They state: “These compounds show significant dose dependent decrease in cell proliferation at higher doses which was more prominent at lower doses in combination with bortezomib (Figure 3G, H).” This is one of several statements in the manuscript where the authors describe significant effects without providing any data regarding statistical significance. Furthermore, in Fig. 3G and 3H, I can see a robust dose-dependent loss of cell numbers upon treatment with Bortezomib, weaker effects for BNC105, and almost no effects for Palbociclib. The reduction of cell numbers after Bortezomib treatment gets only slightly improved when Bortezomib is tested together with BNC105 or Palbociclib. Based on the presented data, it remains unclear whether treating cells with a combination of Bortezomib and BNC105 or Palbociclib leads to a significantly increased reduction of cell numbers compared with Bortezomib treatment (no calculation of IC50 values or statistical significances).

5.     The authors state that they tested the efficacy of Bortezomib and additional proteasome inhibitors in U-CH1 and U-CH2 cells, respectively. IC50 values were supposed to be shown in Tab. 5. Unfortunately, there is no Table 5 in the submitted manuscript.

6.     In the final part of the results section, the authors describe experiments for analyzing the expression of the proteasomal subunits PSMB5 and PSMB8 in U-CH1 and U-CH2 cells by qPCR. They state in the discussion: “We, for the first time, performed the gene expression analysis for proteasomal subunit PSMB5 and PSMB8 and found that these two subunits are increased in chordoma. These findings provide a molecular basis for targeting the proteasomal complex using these proteasomal inhibitors.” This statement is not valid because of several reasons:

(I)         The authors should know that elevated mRNA levels do not necessarily mean that protein concentrations are increased to a comparable extent. It would have been much more meaningful to test PSMB5 and PSMB8 protein levels in chordoma cells in comparison to non-malignant control cells by Western blot. This would have been an easy experiment to perform as high-quality antibodies for both proteins are readily available.

(II)            The authors state that they isolated RNA from U-CH1 and U-CH2 cells, prepared cDNA, and measured the abundance of PSMB5, PSMB8, and GAPDH transcripts by qPCR. They state on page 7 lines 232/233: “We performed the gene expression analysis of proteasomal subunits PSMB5 and PSMB8 by qRT-PCR in U-HD1 and UHD-2 cells as compared to normal human DNA”. It remains unclear what kind of a control “normal human DNA” is supposed to be here. Do they mean DNA or cDNA? What is the source of the material? The caption for Figure 4 adds more to the confusion: “Proteasomal subunits PSMB5 and PSMB8 are increased in chordoma cells U-CH1 and U-CH2. Ct values of A) PSMB5 and B) PSMP8 showing significant Ct value as compared to human DNA (not detected). and C) Ct value of GAPDH. Showing lower Ct values (higher expression) for GAPDH as compared to chordoma cells (U-CH1 and U-CH2).” Why aren’t they able to detect any PSMB5 and PSMB8 transcripts in their control? Why are the Ct values of the GAPDH housekeeping gene massively different between the measurements, and what does this mean for their results? What is a “significant Ct value”?

(III)          The authors do not mention that they DNAse-treated their RNA samples. DNA contaminations can lead to inaccurate results in qPCR-based gene transcript analysis. This can be avoided by choosing qPCR primers that either bind to exon-exon junctions or to different exons that are separated by at least one large intron to prevent the amplification of genomic DNA. However, the primers designed by Ajay et al. either bind to only a single exon (PSMB5: exon 2), or to exons that are interrupted by a relatively short intron (PSMB8: intron between exons 3 and 4 about 400bp, GAPDH: intron between exons 7 and 8 about 200bp). Furthermore, the primers that were chosen for measuring GAPDH expression do not only bind to exons 7/8 of the GAPDH gene, but also to two unrelated loci on chromosomes 6 and X which can lead to the amplification of 101bp long fragments.

Taken together, the results presented for providing evidence that PSMB5 and PSMB8 are upregulated in chordoma cells are highly questionable.

7.     The language and style of all parts of the manuscript require extensive editing. Just a few examples:

(I)             There are already typos in the abstract (the authors misname the U-CH1 and U-CH2 cell lines “U-CHD1” and “U-CHD2”, which are later also called “U-HD1” and “UHD-2” (page 7 line 244)).

(II)          The name of the proteasome inhibitor is Bortezomib and not Bortizomib (Figs. 3G and 3H).

(III)          The captions for most of the figures are confusing (e.g. Fig. 1 “D) Histograms of the mean of average cell count identify any plate pattern.”). See also the caption of Fig. 4.

(IV)         Page 7, lines 211-213: This sentence is a good example of a lot of statements that can be found throughout the whole manuscript that are extremely hard to understand and interpret: “We found three proteasome inhibitors ONX-0914, MG132 and Epoxomicin 211 as hits which demonstrated the reconfirmation but no significant decrease in proliferation of chordoma cells (plotted separately in Figure 3F).” This statement is not further discussed, and I don’t have any idea how the readership should interpret it. How can this be a reconfirmation of the hits when the substances do not lead to a significant decrease in proliferation? Again: How can the authors discuss significant changes here without calculating statistical significance?

(V)           Page 6, lines 195/196: The authors state: “We found 43 compound that were re-confirmed for its activity at the same dose as primary screen (Figure 3).” In Fig. 3, the authors show the dose-dependent inhibition of U-CH1 cell proliferation with 8 different concentrations for each compound. Since the authors do not provide information on the concentrations of the compounds in the primary screen, the meaning of this sentence is unclear.

I find the abundance of mistakes in the manuscript very surprising given that it was approved by 11 authors who are all affiliated with prestigious institutions in the US.

Reviewer 2 Report

Dear Authors,

I don’t know why MDPI gave me this manuscript to review, since I’m a theoretician and my understanding of experiments is low. Maybe because I was modeling proteasome with some ligands.

In my scientific work, I’m looking for papers like yours for inspiration to make modeling. Unfortunately, I did not find it in your paper.

The message from the paper should be clear. We tried hundreds of molecules. The top 10 are:

Place the table with chemical structures and some information on binding affinity. In the supplementary data, you can also place more chemical structures the best and the worst. So theoreticians like me, could have an opportunity to think, why this one works, and that is not.

sincerely

Reviewer 3 Report

The authors present an interesting study looking to identify targets and inhibitors for chordoma. While chordoma is a rare disease, it is one with a poor prognosis, so looking for alternative treatments is essential to improve patient outcomes. The paper is well organized, but there are a few points to address before it can move forward to publication –

1. Introduction does not really give an accurate view of the field particularly line 52-57 - there are several studies and keys bits of data missing. Here are some examples -

(1) PLoS One 8, e78895. doi: 10.1371/journal.pone.0078895 (2013).

(2) Anticancer Res. 34, (2), 623-630. https://pubmed.ncbi.nlm.nih.gov/24510991/ (2014).

(3) J. Pathol. 239, 320-334. doi: 10.1002/path.4729 (2016).

(4) Clin Cancer Res. 2016, 22, 2897-2907. doi: 10.1158/1078-0432.CCR-15-2218 (2016).

(5) Mol. Cancer Ther. 17, 603-613. doi: 10.1158/1535-7163.MCT-17-0324 (2018).

(6) J Med Chem. 62, 4772-4778. doi: 10.1021/acs.jmedchem.9b00350 (2019). 

(7) ChemMedChem. 14, 1693-1700. doi: 10.1002/cmdc.201900428 (2019).

2. Figure 3 should be presented as a table with IC50's in addition to Figure 3

3. The structures of the most potent compounds should be included at some point (a sub-table of hits in the SI at a minimum or a figure of the best 9 or 12 etc in the main tect). The SMILES of the compounds in the plates screened should be added to the supporting information, along with full screening data (U-CH1 and U-CH2)

4. There are alot of typos and little errors that detract from the work, a through proof read is needed. Here are some examples -

- In the abstract 'U-CHD1 and U-CHD2' is written - should it not be 'U-CH1 and U-CH2'

- Line 230 and 233 'U-HD1 and U-UHD2' is mentioned, there are alot of errors here, later line 209 and 237 is it written correctly 'U-CH1 and U-CH2'

- Line 240 remove 'TABLES'

- Line 216 'CKD4/4' should be 'CDK4/6'

- Formatting of the tables needs to be corrected - table 1 and 2 should be moved to the Supporting Information

The formatting of the paper is not quite right, but this can be addressed at the proof stage.

Round 2

Reviewer 1 Report

Ajay et al. have submitted a revised version of their manuscript “High throughput/high content imaging screen identifies proteasomal inhibitors as a therapeutic agent for chordoma”. In the revision, the authors have addressed many of my concerns, however, several central points of their manuscript remain unclear, and I still cannot support its publication.

One main criticism in my first review was that screens with identical cell lines and comparable outcomes (namely that proteasome inhibitors reduce the proliferation of chordoma cells) have already been published and that the manuscript does not provide sufficient novel findings that would justify its publication in “Pharmaceutics”.  In their response, the authors agree with this criticism in most parts, but state that none of the published studies analyzed expression levels of the proteasomal subunits PSMB5 and PSMB8 in chordoma cell lines – which is correct. They claim that the expression of both genes is increased in U-CH1 and U-CH2 cells on mRNA and protein levels and suggest that this overexpression provides a molecular mechanism explaining why proteasome inhibitors repress the growth of chordoma cells. I agree with the authors that such findings would provide novel insights into the biological mechanisms of chordoma cells. However, while I appreciate that the protein expression data have been added upon my request, I still disagree with their statement that the data shown in Fig. 5 provide clear evidence of an upregulated PSMB5 and PSMB8 expression in chordoma cell lines because of the following reasons:

1.    Regarding the RT-qPCR analysis, I asked in my original review what the source of the “normal human DNA” that was used as a baseline to evaluate the levels of PSMB5 and PSMB8 mRNA expression in U-CH1 and U-CH2 cells had been. The authors responded:

“We have modified the sentences for better clarification. We have added the details of the human DNA in the method section. We obtained the human DNA from Millipore Sigma (Cat. no. 11691112001) which was isolated from human blood (buffy coat) and made the cDNA in our laboratory and performed the qRT-PCR. We have modified the Figure 5 (updated figure number) legends.. The Ct values for PSMB5 and PSMB8 were undetectable in normal human cDNA meaning lower expression for both mRNAs. We performed the GAPDH as a reference gene to know that there is enough cDNA. We found Ct values for GAPDH in detectable range in the human cDNA confirming that human samples have enough cDNA as shown by the expression of GAPDH. The Ct values below 30 are considered significant expression in the qRT-PCR studies. The variability of the GAPDH may be because we are comparing different cells.”

Based on this response, it is obvious that all 14 authors who have approved the revised manuscript lack basic knowledge of molecular biology since they tried to synthesize cDNA from genomic DNA, and then used the product of this endeavor to define a baseline in order to show that PSMB5 and PSMB8 mRNA expression is upregulated in U-CH1 and U-CH2 cells. They should know that cDNA is reverse transcribed RNA and that it cannot be synthesized from genomic DNA. It is obvious that the authors do not know how to properly generate, evaluate, and present qRT-PCR gene expression data. At least their statements make it absolutely clear that the mRNA expression data shown in Figs. 5A-C is faulty and does not provide any evidence that PSMB5 and PSMB8 mRNA levels are elevated in chordoma cells in comparison to non-cancer cells.

2.    The authors responded to my request to add PSMB5 and PSMB8 protein expression data to their manuscript since I mentioned that an elevated mRNA expression is not necessarily accompanied by an elevated protein expression:

“Now we have performed western blotting for two chordoma cell lines U-CH1 and U-CH2 and non-cancerous cell lines HEK-293 and HUVEC to confirm our findings that PSMB5 and PSMB8 are therapeutic targets. Thus, these experiments provide incremental but novel insights into the chordoma drug development and biological mechanisms.”

There are several problems with the data shown in Fig. 5D:

1.    In the cited statement shown above, the authors state to have used two non-cancerous cell lines - HEK-293 and HUVEC – to show that PSMB5 and PSMB8 protein levels are elevated in U-CH1 and U-CH2 cells. However, there are no data on HUVEC cells available anywhere in the manuscript.

2.    The quality of the PSMB5 signal in HEK293 cells is very bad and it is almost impossible to compare it with the PSMB5 signals in U-CH1 and U-CH2 cells. The authors do not mention how many times they replicated the protein expression analysis; thus, I assume that these are just the results from a single experiment of low quality. PSMB5 protein expression appears to be lower in the U-CH1 cells than in the HEK-293 cells which is opposite to the mRNA expression data shown. PSMB8 cannot be detected in HEK-293 cells which would have made it even more important to show additional non-cancer controls, since this could be a specific feature of HEK-293 cells.

3.    The bands of the tubulin loading control look completely different than the bands of PSMB5 and PSMB8 and it is highly unlikely that the signals were obtained from the same blot which makes me generally question the results of the protein expression analysis.

4.   Even though they added the protein expression data to their revised manuscript, the authors did not provide any information about this analysis in the methods section which additionally makes interpretation of the data impossible.

The mentioned problems are so substantial that I find any further nuanced discussion of the results to be pointless. The lack of scientific rigor or basic scientific method throughout critical experiments, especially in the revised version of a manuscript, disqualifies this work from publication.

Reviewer 3 Report

The authors need to do a through proofread, once this is done this work is ready for publication.